# Genome-Wide Identification and Expression Analysis of *GATA* Family Genes in *Dimocarpus longan* Lour

**DOI:** 10.3390/ijms25020731

**Published:** 2024-01-05

**Authors:** Kehui Zheng, Jiayue Lu, Xinyu He, Shuoxian Lan, Tingkai Zhai, Shijiang Cao, Yuling Lin

**Affiliations:** 1College of Computer and Information Sciences, Fujian Agriculture and Forestry University, Fuzhou 350002, China; zhkehui@fafu.edu.cn; 2College of Juncao Science and Ecology, Fujian Agriculture and Forestry University, Fuzhou 350002, China; lujiayue231121@163.com; 3College of Forestry, Fujian Agriculture and Forestry University, Fuzhou 350002, China; kachazheduan1117@gmail.com; 4College of Horticulture, Fujian Agriculture and Forestry University, Fuzhou 350002, China; lanshuoxian2023@163.com; 5Institute of Horticultural Biotechnology, Fujian Agriculture and Forestry University, Fuzhou 350002, China; ztk1318347546@163.com

**Keywords:** *GATA* family genes, expression, abiotic stress, embryogenic callus

## Abstract

GATA transcription factors, which are DNA-binding proteins with type IV zinc finger binding domains, have a role in transcriptional regulation in biological organisms. They have an indispensable role in the growth and development of plants, as well as in improvements in their ability to face various environmental stresses. To date, *GATAs* have been identified in many gene families, but the *GATA* gene in longan (*Dimocarpus longan* Lour) has not been studied in previous explorations. Various aspects of genes in the longan *GATA* family, including their identification and classification, the distribution of their positions on chromosomes, their exon/intron structures, a synteny analysis, their expression at different temperatures, concentration of PEG, early developmental stages of somatic embryos and their expression levels in different tissues, and concentrations of exogenous hormones, were investigated in this study. This study showed that the 22 *DlGATAs* could be divided into four subfamilies. There were 10 pairs of homologous *GATA* genes in the synteny analysis of *DlGATA* and *AtGATA*. Four segmental replication motifs and one pair of tandem duplication events were present among the *DlGATA* family members. The *cis*-acting elements located in promoter regions were also found to be enriched with light-responsive elements, which contained related hormone-responsive elements. In somatic embryos, *DlGATA4* is upregulated for expression at the globular embryo (GE) stage. We also found that *DlGATA* expression was strongly up-regulated in roots and stems. The study demonstrated the expression of *DlGATA* under hormone (ABA and IAA) treatments in embryogenic callus of longan. Under ABA treatment, *DlGATA4* was up-regulated and the other *DlGATA* genes did not respond significantly. Moreover, as demonstrated with qRT-PCR, the expression of *DlGATA* genes showed strong up-regulated expression levels under 100 $\mu mol\cdot L^{-1}$ concentration IAA treatment. This experiment further studied these and simulated their possible connections with a drought response mechanism, while correlating them with their expression under PEG treatment. Overall, this experiment explored the *GATA* genes and dug into their evolution, structure, function, and expression profile, thus providing more information for a more in-depth study of the characteristics of the *GATA* family of genes.

## 1. Introduction

A plant’s life begins with the syncytium undergoing seed germination, transitions into the juvenile and mature stages, and then ends with the formation of a new syncytium. During the growth and development process, plants face many influences from the external natural environment, and the results indicate that transcription factors are an important part of a plant’s response to stress [1,2], Recently, many other transcription factors with diverse functions have also been identified, such as the basic leucine zipper [3], MYB (myeloblastosis) [4], NAC [5,6,7,8,9], bHLH (basic helix–loop–helix) [10], ERF (ethylene response factor) [11], CBF (CRT-binding factor) [12], and GATA (GATA-binding factor) [13].

The *GATA* gene family is a transcription factor in eukaryotes, including animals, plants, and fungi [14,15,16,17], and it plays a major part in biological processes, such as developmental differentiation, growth and proliferation, decomposition and apoptosis, and plant responses to environmental transformations. So far, GATA transcription factors have been found in many plants, such as rice [18], *Arabidopsis thaliana* [16], *Pyrus bretschneideri* [19], *tobacco* [20], *Black Soybean* [21], and *tomato* [22]. Within the context of plant hormone signaling pathways, the *GATA* gene family is important, particularly in hormone synthesis, signal transduction, and responsiveness. For instance, GATA transcription factors can modulate gibberellin biosynthesis and degradation, thereby influencing plant growth and development. Additionally, *GATA* genes are involved in the signal transduction and response of plant hormones, such as abscisic acid, ethylene, and ABA. In summary, *GATA* genes have diverse functions in plant hormone regulation. GATA transcription factors have a highly conserved class IV zinc finger structure and can recognize and specifically bind DNA sequences (T/A) (GATA(A/G)) [18,23]. Most *GATA* proteins contain CX2CX17–20CX2C zinc finger domains [24], followed by a basic region [19,25,26]. In addition, 30 and 28 gene members of the *GATA* family have been identified in *Arabidopsis (Arabidopsis thaliana)* [24] and rice (*Oryza sativa subsp. japonica*) [24], respectively. However, the complete genome of *DlGATA* has not yet been characterized in *Dimocarpus longan*.

Longan originates from Fujian, Guangdong, Southeast Asia, and it is also cultivated in Yunnan [27,28]. It has a long history of cultivation [29]. Therefore, it occupies an important position in the study of fruit trees. At the same time, a series of biotechnologies related to exogenous plant hormones also play an active role in overcoming harsh environments and genetic limitations, as well as improving crop quality and storage conditions [30,31]. Therefore, the study of plant hormones plays an indispensable role in understanding the effects of many plants, and this is particularly groundbreaking in the case of longan, which is the subject of the experiments described here. In this study, bioinformatics was used to analyze and identify members of the *DlGATA* family at the genome-wide level; this was mainly for the identification of *DlGATA* genes and their basic physicochemical properties, chromosomal localization, phylogenetic tree, gene structure, cis-regulatory elements, and expression patterns. In addition, the effects of treatments with the exogenous plant hormones IAA and ABA on the expression of *GATA* genes in early embryogenic callus of longan were also analyzed using real-time quantitative PCR. The study of the *DlGATA* family is conducive to gaining a deeper understanding of the *GATA* family and its structure and function, as well as the promotion of in-depth experiments.

## 2. Results

### 2.1. Characterization of the DlGATA Gene in Longan and Its Positional Distribution on Chromosomes

A total of 22 *GATAs* were identified among the longan genes. It was found that 21 *GATAs* were distributed on nine different chromosomes, and 1 was a *GATA* gene that was not mapped on the fixed chromosomes. They were renamed *DlGATA1*–*DlGATA22* according to their positions (Figure 1). Among these, chr1 was the longest and possessed the most *DlGATA*s (5 (23.8%)), and chr15 was the shortest. On chromosomes chr12 and chr15, only one *GATA* gene (4.8%) was found. To investigate the characteristics of *DlGATA*, its physicochemical properties were analyzed (Table 1). It was found that in *DlGATA1*–*DlGATA22*, the protein-encoded amino acid numbers ranged from 112 to 542, the relative molecular weights ranged from 12,336.41 to 60,405.08, and the lengths of the proteins showed a close positive correlation with the relative molecular weights. Regarding the physicochemical properties of the theoretical isoelectric point, 10 *DlGATAs* were acidic, and the other 12 were alkaline. The highest isoelectric point for both *DlGATA6* and *DlGATA22* was 10.89. Except for the slightly lower stability index of *DlGATA9*, all GATA transcription factors had an instability index greater than 40, making them more prone to denaturation, aggregation, and sedimentation. Their aliphatic coefficients were close together, except for that of *DlGATA16*. All GRAVY values were less than 0, which indicated that all of the *DlGATA* proteins were hydrophilic. In terms of their subcellular localization, all *DlGATAs* were located in the nucleus.

### 2.2. Phylogenetic Analysis between Longan, Apple, and Arabidopsis

The construction of phylogenetic trees is beneficial for the study of the biological relationships and genetic relationships of the *GATA* transcription factor gene family and other gene families. Using the MEGA X software, a biological evolutionary tree consisting of 35 *MdGATA* proteins, 22 *DlGATA* proteins, and 30 *AtGATA* proteins was obtained (Figure 2). According to the cluster analysis, we found that longan was the same as *Arabidopsis* and apple in that it could be categorized into four subfamilies; of these, subfamily II contained the most *GATA* genes, as it had 11 *DlGATA* proteins. It was followed by subfamily IV, which had five *DlGATA* proteins, namely, *DlGATA1*, *DlGATA3*, *DlGATA5*, *DlGATA11*, and *DlGATA16.* Subfamily Ⅰ was the smallest and only had two *GATA* proteins. Through the study of the *DlGATA* proteins, we aimed to gain a clearer understanding of the evolutionary process and possible connections with the evolution of other species.

### 2.3. Analysis of the Gene Structure and Conserved Protein Sequence of DlGATA

In order to gain a clearer understanding of the gene structure of *DlGATA* and its protein sequence, we analyzed its exon–intron structure, its protein motifs, and its conserved domain structure. The results in Figure 3B show that some independent motifs appeared in only one family. For example, motif4 only existed in subfamily II, while motif5 and motif3 were only present in subfamily III. In subfamily III, motif5 was located in the upstream region of motif3 and motif1, and motif3 was upstream of motif1. Furthermore, motif1 was in all of the *DlGATA*s, suggesting that motif1 was a very important conserved domain within the *DlGATA* genes. In the exon/intron structure (Figure 3D), it was found that 2 to 3 exons were generally present in subfamily III, while 7 to 11 exons were present in subfamily III. The UTR region was not present in subfamily I. The present study was carried out to offer an essential basis to reveal the structural characteristics of longan.

### 2.4. Synteny Analysis of Longan and Arabidopsis Genes and an Analysis of Replication Events within Gene Families

To understand the evolutionary mechanism of *DlGATA* and study its possible evolutionary links to other species, we analyzed the synteny of the *DlGATA* genome with the *AtGATA* genome (Figure 4). The results indicated that there were nine pairs with collinearity between *DlGATA* and *AtGATA*, with the largest number of collinearities being in chromosome 6 (three pairs). Two collinear pairs existed in chromosome 7, and one collinear gene pair between the *DlGATA* genome and *AtGATA* genome was found in chromosomes 5, 10, 13, and 14.

As can be seen in Figure 5, it was also found that there were five pairs of *GATA* genes in the longan chromosome, namely, *DlGATA6*/*DlGATA11*, *DlGATA6*/*DlGATA5*, *DlGATA15*/*DlGATA17*, *DlGATA18*/*DlGATA13*, and *DlGATA1*/*DlGATA3*. The first four pairs were segmental duplication events in *DlGATA* genes, and the last pair was a tandem duplication event.

### 2.5. Analysis of the Cis-Acting Elements in the Promoter of the Longan GATA Genes

By analyzing the *cis*-acting elements, it was found that a total of 46 types of *cis*-acting elements existed in the promoter region of the *GATA* gene of longan (Figure 6), such as ABE, ABER, SARE, BOX4, BOX11, and so on. All of these *cis*-acting elements were divided into four types, namely, stress responses, hormone responses, plant growth and development, and light responses. Among the many *cis*-acting elements, Box 4 was the most widely distributed among the 22 *DlGATA*s*;* its number was 75 in total, and it was followed by ABE with 64. On the whole, the numbers of stress responses and light responses were significantly greater than those of hormone responses and plant growth and development elements. The results showed that the longan *GATA* family may be more sensitive to stress and light responses.

### 2.6. Expression Analysis of Longan at Different Temperatures, Concentration of PEG, and Early Developmental Stages of Somatic Embryos

By using data from the longan database, the gene expression of *DlGATA* at different temperatures, concentrations of PEG, and early developmental stages of somatic embryos was obtained, and a heat map was made (Figure 7A and Appendix A). In the temperature expression section of *DlGATA*, the results showed that the *GATA* gene expression was roughly divided into three categories. In the first category, the most significant up-regulated expression at 15 °C occurred for *DlGATA8*, *DlGATA13*, *DlGATA21*, *DlGATA14*, *DlGATA1*, and *DlGATA2*. In the second category, the most significant up-regulated expression at 25 °C occurred for *DlGATA9*, *DlGATA4*, *DlGATA6*, *DlGATA16*, and *DlGATA19*. The third category included the last five *GATA* genes shown in the figure, which had a stronger up-regulated expression at 35 °C than at the other temperatures. Therefore, we hypothesized that the first group of genes contributed more to cold resistance in longan, while the third group of genes was important in the mechanism of heat tolerance in longan.

At different concentrations of PEG (5%, 7.5%) conditions in the *DlGATA* expression fraction, the following results were obtained (Figure 7B and Appendix A): the *GATA* gene showed a strong up-regulated expression level under all PEG concentration conditions. Among them, *DlGATA2*, *DlGATA8*, *DlGATA13*, *DlGATA15* and *DlGATA21* had strong up-regulated expression levels in the absence of PEG. *DlGATA9* and *DlGATA17* showed high up-regulated levels of expression under 5% PEG. In addition to that, the *DlGATAs* showed a significantly up-regulated expression profile at 7.5% PEG.

The following types of expressions exist for *DlGATA* in the heatmap section during early developmental stages of somatic embryos (EC: embryogenic callus, ICpEC: incomplete pro-embryogenic callus, GE: globular embryo) (Figure 7C and Appendix A) First, there was a continuous down-regulation in the expression from EC to ICpEC in the GE stage, and this included *DlGATA12*, *DlGATA1*, and *DlGATA13*. Second, expression was unchanged in these three stages, and this scenario included *DlGATA19*, *DlGATA3*, and *DlGATA11*. Third, there was an upward modulation during the EC and ICpEC phases and a downward modulation during the GE phase for *DlGATA15*, *DlGATA5*, *DlGATA20*, *DlGATA2*, and *DlGATA8*. Fourth, four genes—*DlGATA21*, *DlGATA14*, *DlGATA18*, and *DlGATA9*—were downregulated in EC and upwardly modulated in IcpEC and GE. Fifth, *DlGATA10*, *DlGATA4*, and *DlGATA16* were upwardly modulated in EC and GE and downregulated in ICpEC. Lastly, *DlGATA6*, *DlGATA22*, *DlGATA7*, and *DlGATA17* were downregulated in the EC and ICpEC stages and upwardly modulated in the GE stage.

To study the expression of *DlGATA* during the early developmental stages of longan somatic embryos in greater depth, five representative *DlGATAs* were selected for qRT-PCR (Figure 8). The following profiles were found: the four genes other than *DlGATA1* had strong profiles in the GE phase, indicating that *DlGATA* may promote the differentiation of GE.

### 2.7. DlGATA Expression Profiles in Nine Different Tissues

The expression of *DlGATA*s was further explored using RNA-seq data from flowers, flowerbuds, leaves, pericarps, pulps, roots, seeds, stems, and young fruits. According to the expression conditions, the *DlGATA* expression levels were used to make a heatmap (Figure 9). On the whole, the different *DlGATA*s had different tissue expression levels, and high levels of expression were likely to be beneficial. In the current study, *DlGATA12*, *DlGATA7*, and *DlGATA9* were highly expressed in the flowers and flowerbuds of longan, indicating that they may have a promotive role in the germination processes of flowers. *DlGATA4*, *DlGATA11*, *DlGATA10*, and *DlGATA21* showed strong up-regulated expression in the root, which may have a driving effect on root extension. However, in the expression of *DlGATA*, not all expression levels were completely different, and a small number of genes had similar expressions. For example, *DlGATA17*, *DlGATA14*, and *DlGATA20* had similar expressions.

### 2.8. Expression of DlGATA Induced by Exogenous Hormone Treatments

According to the heatmap of *DlGATA* expression in the early somatic embryo (Figure 7), 5 representative *GATA* genes with different expression patterns were selected from the 22 *GATA* genes; these were *DlGATA1*, *DlGATA4*, *DlGATA12*, *DlGATA14*, and *DlGATA17* (Figure 10). The data showed that, as a whole, there was a significant expression profile with IAA for all five genes, and there was a less significant one with ABA. *DlGATA1* showed the most significant expression under IAA stress, as it was nearly five-times greater than that in the control group. In addition, the expression of *DlGATA* at different concentrations of IAA stress was analyzed in this experiment. The findings demonstrated that the five genes (*DlGATA1*, *DlGATA4*, *DlGATA12*, *DlGATA14*, and *DlGATA17*) were significantly up-regulated, expressed at a concentration of 100 $\mu mol\cdot L^{-1}$. At the same time, some of the *DlGATA*s, such as *DlGATA1* and *DlGATA4*, were greatly expressed at higher concentrations. However, at lower concentrations of the IAA treatment, the genes in the experimental group were mostly suppressed or maintained at expression levels similar to those in the untreated group. The results indicated that IAA at a concentration of 100 $\mu mol\cdot L^{-1}$ induced *DlGATA* expression, while concentrations too high or too low may inhibit this expression.

## 3. Discussion

In this study, the longan *GATA* gene family was divided using bioinformatic technology. A total of 22 *DlGATA* genes exist, and they can be divided into four families. Of these, subfamily II has the most members, while subfamily I has the fewest *DlGATA*s. In the assay of the physicochemical properties of *DlGATA*, it was found that *DlGATA* proteins and *AcoDREB* proteins [32] had similar isoelectric points, mostly centered on the 5 to 9 isoelectric points, and there was no obvious acidic versus basic nature overall. In the analysis of the structure of *DlGATA* and its protein conservation sequence (Figure 3), it was shown that the gene structures and protein conservation sequences in the same subfamily were roughly similar, and the structures of members of different subfamilies were different from the conserved domains of proteins [33]. This was in line with previous results. Furthermore, the number of introns has a key influence on the evolution and generation of plant gene families [34]. The intron number varied in the same *GATA* gene family in longan. The protein–protein role was closely linked to the motif composition of the transfer factors [35]. The motif combinations present in the GRAS family of *Arabidopsis* can function as transcriptional regulatory proteins to mediate many different interactions with the basic transcriptional machinery and accessory proteins [36]. In the motif study of longan *GATA*, only motif3 and motif5 were present in subfamily III, so it can be speculated that motif3 and motif5 are an important basis for its recognition, and this is most likely the reason for the specific functions carried out by subfamily III and its accessory proteins.

In the process of phytohormone signaling, related genes are substantially enriched [37,38]. Studies have revealed that phytohormones are instrumental in EC regeneration [39,40,41]. There are many singular genes related to embryonic growth and development in the EC of longan [42]. In the present study of the embryogenic callus of longan, it was found that growth hormones significantly induced the expression of *DlGATA* genes. This was in line with the findings of a previous study of longan development, which showed that there were large numbers of auxin and cytokinin signaling components in the transcriptomes of the four different stages of NEC (non-embryogenic callus), ICPEC, EC, and GC [37]. This suggests that *DlGATA* may synergize with IAA and participate in biotic or abiotic stresses associated with it. In addition, *DlGATA1*, which was significantly expressed under growth hormone treatment, showed strong expression levels in the stems and young fruits. In previous studies, it was shown that an elevated ratio of auxin to cytokinin favors the rooting of point–stem cuttings of rose shoots [43]. It was also reported that the concentration of auxin in Chinese fir stems affected the activity of cambium [44]. Studies have shown that increased concentrations of IAA reduce the sensitivity to exogenous material in ripe fruits of *Citrus sinensis* (L.) *Osb* [45]. It can be speculated that *DlGATA* may be involved in the stem elongation mechanism and fruit anti-shedding mechanism based on IAA signaling. However, the expression levels of *DlGATA* in various tissues under IAA treatment need to be further explored, as this will provide a rational foundation for further study of the growth and development of longan, as well as the function of *DlGATA*.

The effect of drought on longan was found to contribute to higher yields and higher economic returns. The *GATA* showed a close association with drought stress [14,46,47]. In a study of wheat *GATA*, we found that ABA with Ca^2+^ caused the up-regulation of T*aGATA*s, which activated related genes [48]. In studies of abiotic stress in grain crops such as rice and soybean, ABF, a response factor to ABA, has been shown to bind to ABER and promote the activation of related genes upon receipt of a signal [49]. In a study of the promoter region of *DlGATA*s, we found that ABER, a *cis*-acting element associated with the hormone ABA, was abundantly present in the promoter region of *DlGATA*s. Interestingly, we did not find a significant correlation with ABA in our analysis of *DlGATA* expression in longan, and this could be a result of the repressive effect of silencers in gene expression or other reasons, making it different from *GATA* genes in other species [50]. Furthermore, in the study of the IAA hormone, we found that *DlGATA4*, *DlGATA10*, *DlGATA11*, and *DlGATA21* were significantly expressed in roots. Similarly, strong expression levels were also shown in the presence of IAA at a concentration of 100 $\mu mol\cdot L^{-1}$. In previous studies, *Arabidopsis* PHB3 was found to regulate the degradation of IAA14/28 through the non-mediated growth hormone signaling pathway by releasing ARFs to bind to *GATA23* and modulating the initiation of lateral root primordia [51]. During this experiment, we hypothesized that IAA may promote the transcription of *DlGATA*s through a series of unknown signaling mechanisms (Figure 11), which, in turn, modulate root traits. Changes in root traits can result in a better response to drought-stressed environments [52,53,54,55,56]. Therefore, we hypothesized that the drought response mechanism of longan *GATA* is less associated with ABA, while there is a strong link with IAA. In addition, to further investigate the role of *DlGATA* under drought conditions and the expression levels under different drought stresses, we conducted further studies on this. It was found that *DlGATA1*, *DlGATA4*, *DlGATA12*, and *DlGATA14*, which were mentioned in previous experiments (Figure 7), were also strongly up-regulated, expressed under 7.5% PEG at different concentrations of PEG. Meanwhile, *DlGATA17* also showed strongly up-regulated expression at 5% PEG, further strengthening our speculation of *DlGATA* under drought conditions. Based on this, it is possible to predict the role and function of *DlGATA* in the drought resistance process.

## 4. Materials and Methods

### 4.1. Plant Materials

‘Honghezi’ longan used in this study was provided by the Institute of Horticultural Biotechnology of Fujian Agriculture and Forestry University, which grows mainly in Fuzhou, Fujian, and it has a high yield and cannot easily drop fruits; the climatic conditions required for growth are sunny, favorable temperatures, and high-humidity areas. The longan *GATA* gene sequences, CDS sequences, amino acid sequences, and gene annotation information were downloaded from the longan genome database, which was constructed in our laboratory [59].

### 4.2. Identification of GATA Genes in Longan

The reported *Arabidopsis GATA* gene family sequences were obtained from the TAIR [60] (https://www.arabidopsis.org/ (accessed on 9 July 2023)) website and homologated to the longan genome database using the TBtools V2.012 [61] software (Chen Chengjie from South China Agricultural University, China). The Interpro website [62] (https://www.ebi.ac.uk/interpro/ (accessed on 9 July 2023)) was used to view the conserved structural domains of the above sequences, and filtering based on whether or not the zinc finger structure was included (pfam00320) ultimately yielded 22 sequences of the longan *GATA* family. Referring to the longan genome annotation file and the nomenclature of the *GATAs* of other species, they were sequentially named *DlGATA1*–*22*. The number of amino acids, relative molecular weight, theoretical isoelectric point, instability index, aliphatic index, and grand average of hydropathicity of the *DlGATA* family members were analyzed using the Expasy online software [63] (https://www.expasy.org/ (accessed on 20 July 2023)) and the cello online software [64] (http://cello.life.nctu.edu.tw/ (accessed on 20 July 2023)) for the subcellular localization of the *DlGATA* family.

### 4.3. Phylogenetic Analysis, Gene Structure, and Conserved Motif Analysis

The MEGA X [65] software (Mega Limited, Auckland, New Zealand) was used to construct the phylogenetic tree of the *GATA* family for three species—longan, *Malus domestica*, and the model plant *Arabidopsis thaliana*. The *GATA* sequences of apple and *Arabidopsis thaliana* were obtained from the website of PlantTFDB [66] (http://planttfdb.gao-lab.org/ (accessed on 6 December 2022)), and the maximum likelihood was chosen to compute these sequences with 1000 bootstrap replicates. Finally, the above results were embellished using the iTOL online website [67]. The longan *GATA* sequences were imported into the MEME online website [68] (https://meme-suite.org/meme/ (accessed on 9 July 2023)), and the number of motifs was set to 10 to analyze the prediction of the above sequences. The structural domains contained in the amino acid sequence of *GATA* of longan were analyzed using the Batch CD-search function on the NCBI website (https://www.ncbi.nlm.nih.gov/ (accessed on 9 July 2023)). The longan *GATA* gene annotation files were imported into the Tbtools software [61]. Based on the Gene Structure View function in this software, the results of the above three types of analysis were visualized, and, finally, maps of the longan *GATA* motifs, introns, and conserved structures were acquired.

### 4.4. Chromosomal Location, Cis-Acting Element Analyses, and Analysis of Collinearity with Other Species

PlantCARE [69] (https://bioinformatics.psb.ugent.be/webtools/plantcare/html/ (accessed on 16 July 2023)) was utilized to analyze the *cis*-acting elements of the 2000 bp sequence upstream of *GATA* in longan. Finally, a two-dimensional heatmap was drawn, with the components involved in stress responses, hormone responses, plant growth and developmental regulation, and light responses as horizontal coordinates and the longan *GATA* genes as vertical coordinates. The genome-wide annotation information of *Arabidopsis thaliana* and rice was downloaded from the TAIR website and the Phytozomehttps website [70] (https://phytozome-next.jgi.doe.gov/ (accessed on 16 July 2023)), respectively, and analyses of the synteny of these species with longan were performed using the McscanX (https://github.com/wyp1125/MCScanX/ (accessed on 16 July 2023)) software (Tang Haibao from Fujian Agriculture and Forestry University, Fuzhou, China) [71]. Finally, Tbtools was used to draw and enhance the visualized images of the gene synteny among longan, *Arabidopsis*, and rice. The collinear relationships of the *GATA* gene within the longan species were analyzed using the MCscanX software, and the results were visualized using TBtools.

### 4.5. Hormonal Processing of Longan EC, RNA Extraction, and qRT-PCR Analysis

Longan embryogenic callus materials with a good growth state were selected, and equal amounts were added to a liquid medium with 50 $\mu mol\cdot L^{-1}$, 100 $\mu mol\cdot L^{-1}$, 200 $\mu mol\cdot L^{-1}$ IAA, and 100 $\mu mol\cdot L^{-1}$ ABA; longan EC was added to a liquid medium containing MS without any hormones as a blank control, and three replicates were set for the above treatments; they underwent 24 h of incubation at 25 °C in the dark, and they were then filter-dried and frozen at −80 °C in a refrigerator for later use. The DNAMAN6 software (Lynnon Biosoft, San Ramon, CA, USA) was used to design the qRT-PCR primers for the longan *GATA* gene, and the primer sequences are shown in Appendix A. RNA was extracted from the above hormone-treated materials using the TransZol kit, as described in the instructions. The RNA obtained through extraction was reverse-transcribed into cDNA using the Revertaid Master Mix (Thermo Fisher Scientific, Shanghai, China) kit. By using ubiquitin [72] (UBQ) as an internal control, the expression level of *GATA* in longan after the IAA and ABA hormone treatments and the somatic embryo of longan (EC: embryogenic callus, IcpEC: Incomplete pro-embryogenic callus, GE: globular embryo) stage was examined in a Roche Light Cycler 96 (place of origin: Basel, Switzerland) for qRT-PCR. The 2^−ΔΔCt^ method was used to calculate the assay data, then averaged over three repetitions and analyze the significant differences in the relative expression of *DlGATA* at the three stages of longan somatic embryo using the software Prism 8.0.2 as well as the one-way ANOVA method. The relative expression of *DlGATA* after IAA and ABA treatments was analyzed using SPSS 25 as well as Duncan’s one-way ANOVA method, and, finally, Prism 8.0.2 was used to visualize the above results. Three repetitions were used, and the average value was taken. The statistical method used in this experiment was standard deviation.

### 4.6. Analysis of the Specific Expression of DlGATA Family Genes

The FPKM of longan *GATA* family members at three stages of longan’s somatic embryogenesis (EC, IcpEC, GE) at high and low temperatures (35 °C, 15 °C), PEG (5%, 7%), 2,4-D treatments and at nine different tissue sites (seeds, roots, stems, leaves, flowers, flowerbuds, pulp, young fruits, and pericarp) was derived from the transcriptome database (SRA050205). The above data were log-transformed (${log}_{2}$) and plotted on a heatmap using Tbtools.

## 5. Conclusions

In the present experiment, based on a heatmap of expression in early embryonic calluses of longan, 5 representative genes extracted from 22 *DlGATA* genes were investigated. Based on the phylogeny and structure of *GATA*, its evolutionary relationship with *Arabidopsis* and apple was determined. In the analysis of Collinearity within the longan, there was one tandem gene pair and four fragment duplication gene pairs in the *GATA* gene family that provide information for studying gene amplification. Under the expression of *DlGATA* at different concentrations of IAA, they were all found to have a strong up-regulated expression profile under IAA conditions at 100 $\mu mol\cdot L^{-1}$ concentration. We also found that *DlGATA* was strongly up-regulated, expressed in roots and stems, which we correlated with drought stress. In addition, we investigated the expression of *DlGATA* under different concentrations of PEG. Further information about the relationship between *DlGATA* and drought was obtained. In addition, further studies of expression in the somatic embryo of longan can offer a rationale for exploring the functions and actions of *GATA* in the early period of development. All of these can help us to improve our understanding of *DlGATA*. However, there are still limitations to the in-depth study of the *GATA* gene family, so there are still questions that need to be answered, making this another fruitful area with room for progress.

## Figures and Tables

**Figure 1 ijms-25-00731-f001:**
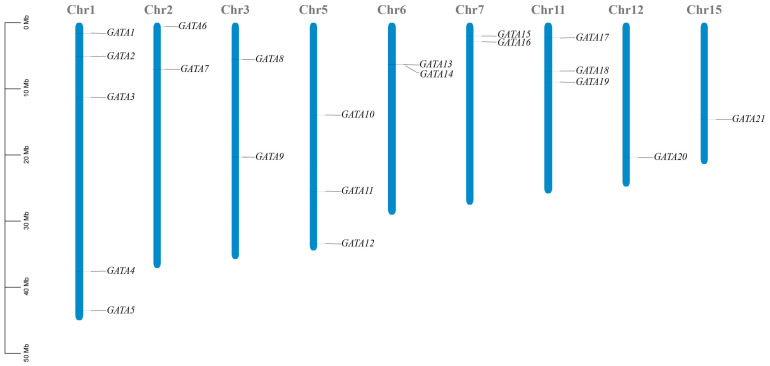
Distribution of the *DlGATA* gene locations on the chromosomes of longan. The measurement bar on the left shows the relative position and the length of the chromosome in Mb.

**Figure 2 ijms-25-00731-f002:**
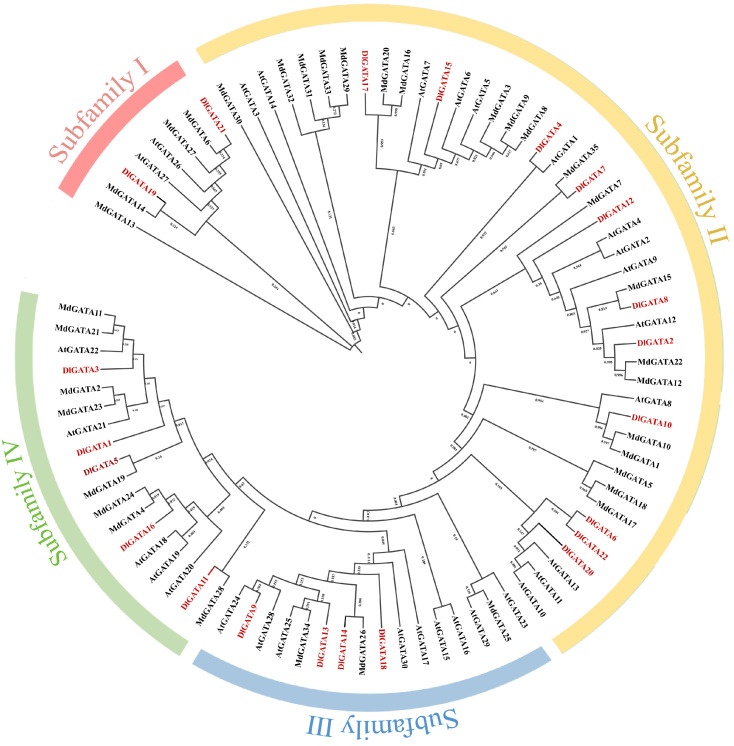
Phylogenetic tree of longan, *Arabidopsis* and *apple*. Different colors were used to represent subfamilies I, II, III and IV, consisting of 35 *MdGATAs*, 22 *DlGATAs* and 30 *AtGATAs*.

**Figure 3 ijms-25-00731-f003:**
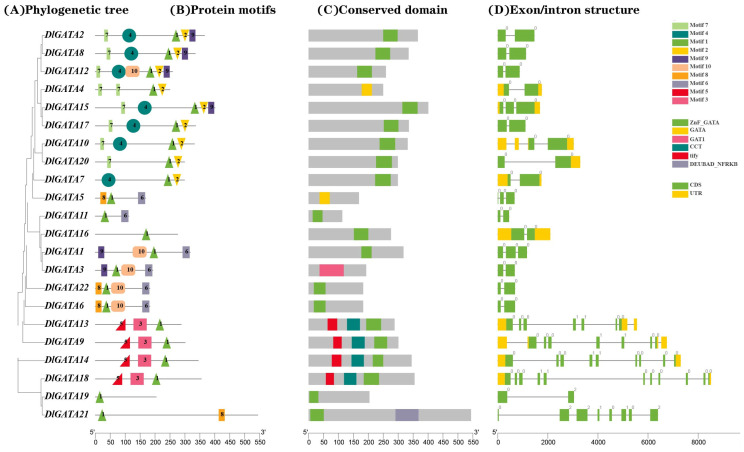
Schematic representation of the phylogenetic relationships of *GATA* and a gene distribution diagram. (**A**) The phylogenetic tree of *DlGATA* was constructed using the MEGA X software, and the results were consistent with those obtained using the maximum similarity method with 1000 repetitions. (**B**) The positional distribution of protein motifs. (**C**) The conserved domain of the *DlGATA* proteins. (**D**) The exon/intron structure of *DlGATA*; standard green modules represent exons, grey lines indicate introns, and standard yellow modules represent UTR regions.

**Figure 4 ijms-25-00731-f004:**
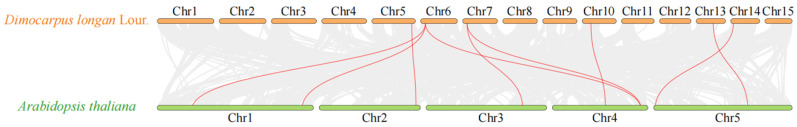
Synteny analysis between longan and *Arabidopsis* genes. Red lines are collinear with *GATA* in two plants and gray indicates collinear gene pairs in longan and *Arabidopsis* genes.

**Figure 5 ijms-25-00731-f005:**
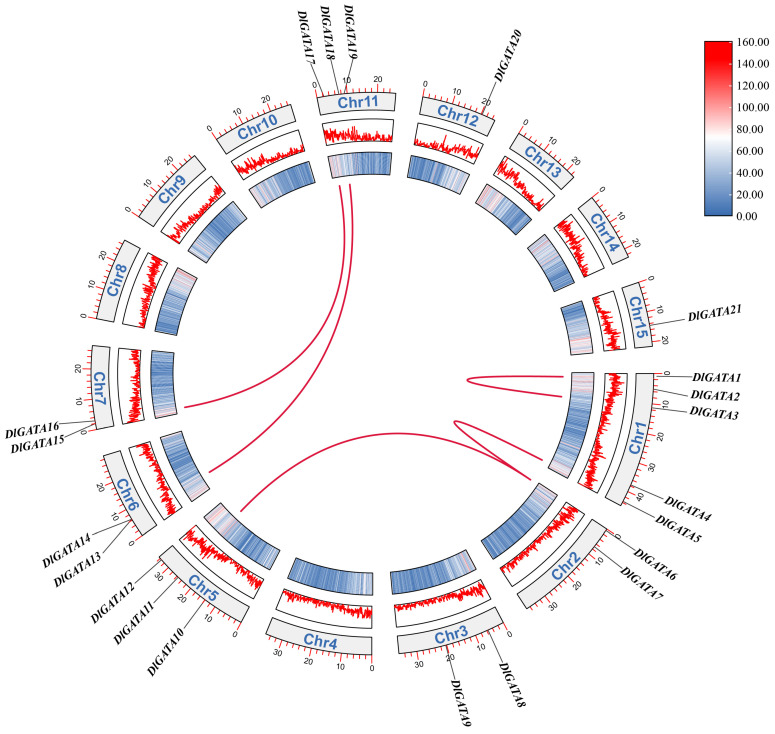
Collinearity analysis of the family of *GATA*s in the longan chromosomes. Red lines indicate homologous gene pairs in which segmental duplications were present. The gray curved frames represent the different chromosomes of longan.

**Figure 6 ijms-25-00731-f006:**
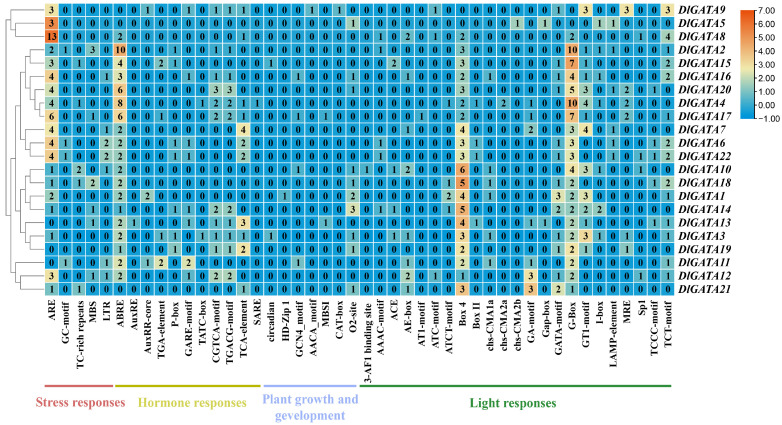
Schematic distribution of *cis*-acting element positions in each *DlGATA*. Plot of the positional distribution of the specific numbers of 46 cis-acting elements identified in the 22 *DlGATA*s.

**Figure 7 ijms-25-00731-f007:**
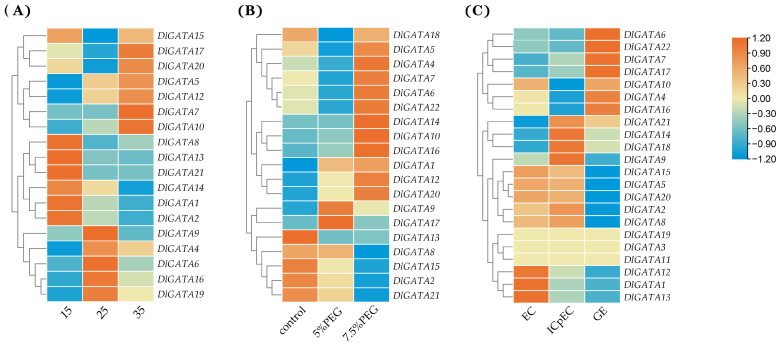
Heat map of *DlGATA* expression in longan at different temperatures, PEG concentrations and early developmental stages of somatic embryos. (**A**) Expression of *DlGATA* under temperature treatments (15 °C, 25 °C, 35 °C). (**B**) Expression of *DlGATA* under PEG treatment (0%, 5%, 7.5%). (**C**) Expression of *DlGATA* in three stages of early somatic embryo development.

**Figure 8 ijms-25-00731-f008:**
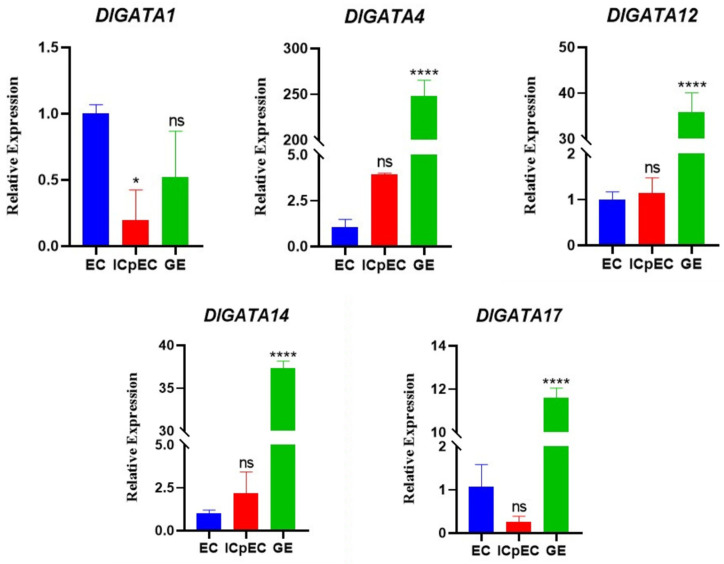
Expression levels of related genes in different early developmental stages of somatic embryos. Analyze the significant differences in the relative expression of *DlGATA* at the three stages of longan somatic embryo using the software Prism 8.0.2 as well as the one-way ANOVA method. Three experiments were repeated and their average values were taken. The statistical method used in this experiment was standard deviation. (* *p* < 0.1, **** *p* < 0.0001).

**Figure 9 ijms-25-00731-f009:**
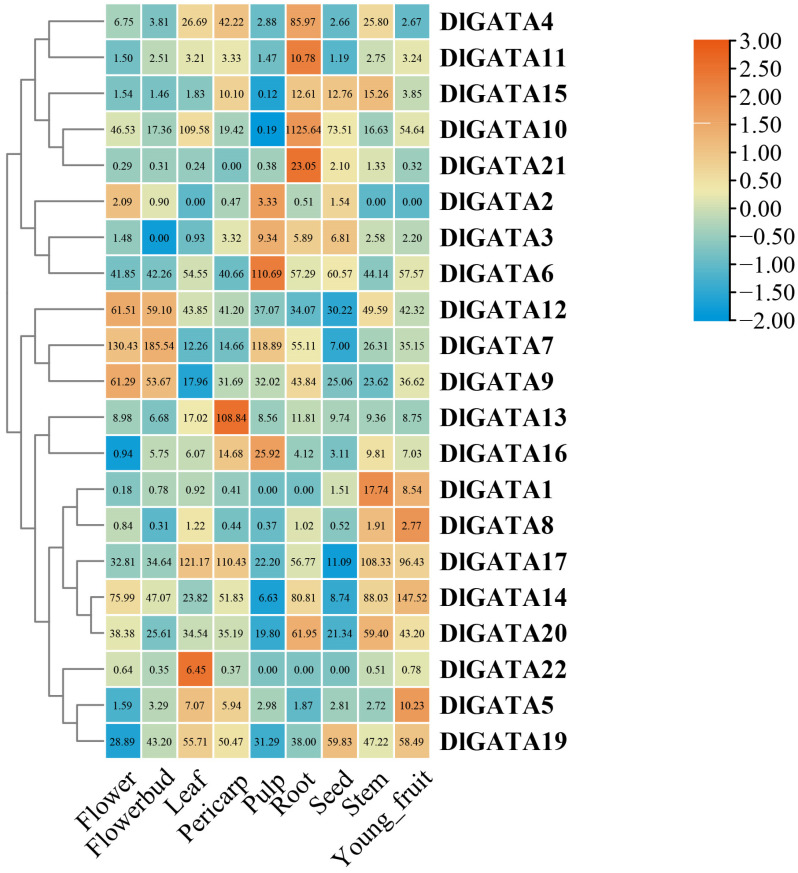
Heatmap of the expression of the longan *GATA* gene in nine different tissues. Red represents higher relative expression levels, and blue indicates lower relative expression levels.

**Figure 10 ijms-25-00731-f010:**
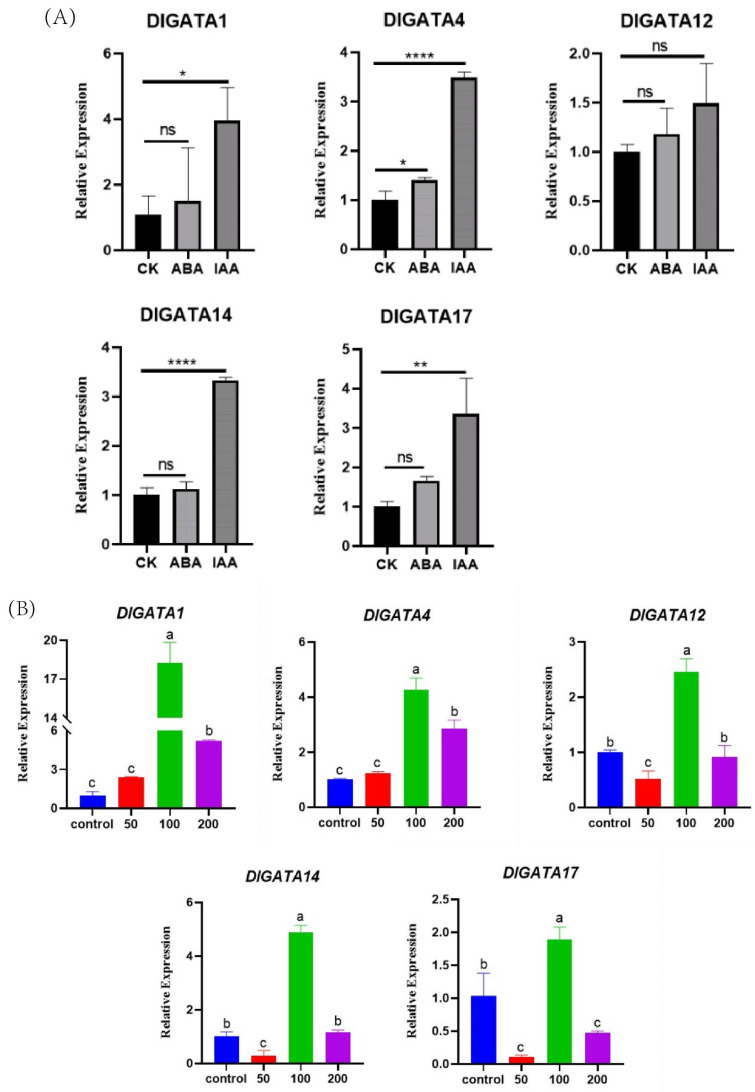
The *GATA* gene expression in early somatic embryo longan was analyzed using qRT-PCR under stress due to different concentrations of the exogenous phytohormones IAA and ABA. (**A**) Expression of related genes under IAA and ABA stress at a concentration of 100 $\mu mol\cdot L^{-1}$. (* *p* < 0.1, ** *p* < 0.01, **** *p* < 0.0001). (**B**) Expression of related genes treated with IAA stress at concentrations of 50 $\mu mol\cdot L^{-1}$, 100 $\mu mol\cdot L^{-1}$ and 200 $\mu mol\cdot L^{-1}$. The letters at the top of this bar graph: the same letter means there is no significant difference, different means there is a significant difference.

**Figure 11 ijms-25-00731-f011:**
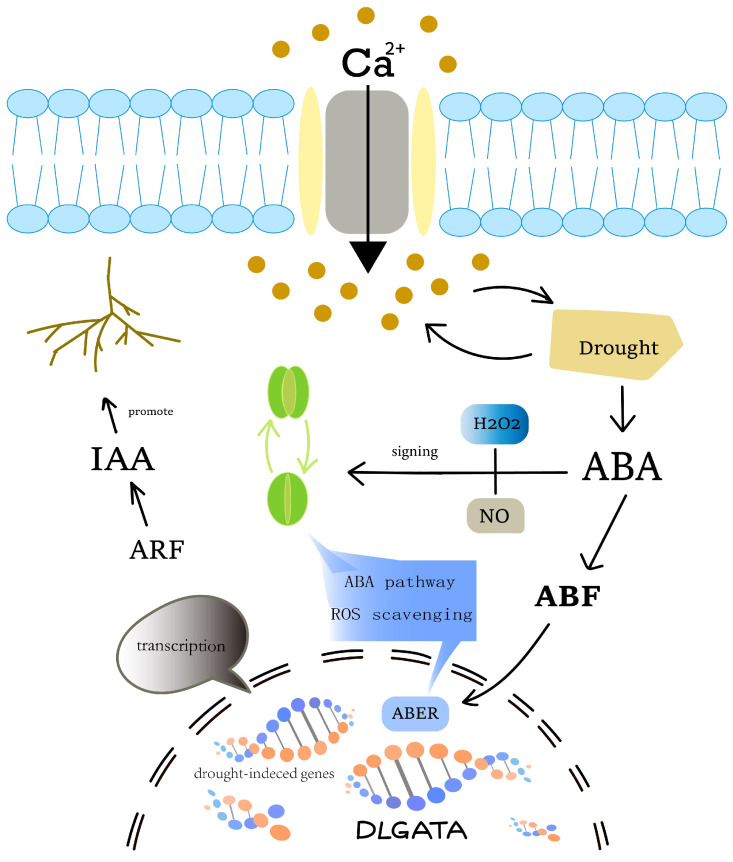
Diagram of signaling of *GATA* genes in longan under drought conditions. Under drought conditions, the outside plant produces high stress, allowing Ca^2+^ to enter cells through ion channels. Both Ca^2+^ and drought stress induce an increase in ABA hormones. ABF acts as a response-binding factor for ABA and binds to the *cis*-acting element ABER located on *DlGATA* upon the receipt of a signal, but its expression is inhibited for other mechanisms, such as its internal silencers. Therefore, the ABA signal and its secondary messengers H_2_O_2_ and NO [57,58] may not directly cooperate with *GATA*. In the IAA response mechanism, drought stimulation leads to changes in the concentration of IAA, and signaling occurs in the IAA response factors—ARFs—which induce related genes and contribute to inter-root growth [51].

**Table 1 ijms-25-00731-t001:** Physicochemical characterization of *GATA* family genes in longan.

Group	Gene Accession	Gene Id	Size/aa ^1^	MW ^2^/Da	Theoretical pI ^3^	Instability Index	Aliphatic Index	GRAVY ^4^	Subcellular Localization
I	Dlo023940	*DlGATA19*	203	22,746.48	9.41	58.31	64.38	−0.855	Nuclear
	Dlo031787	*DlGATA21*	542	60,405.08	6.45	53.84	67.91	−0.652	Nuclear
II	Dlo000544	*DlGATA2*	365	40,147.31	5.75	50.23	56.41	−0.715	Nuclear
	Dlo002694	*DlGATA4*	249	27,832.37	8.81	48.32	59.08	−0.642	Nuclear
	Dlo003563	*DlGATA6*	182	19,817.05	10.89	62.74	67.58	−0.513	Nuclear
	Dlo004225	*DlGATA7*	298	32,967.26	8.07	56.55	67.01	−0.606	Nuclear
	Dol006515	*DlGATA8*	334	36,529.19	6.26	45.76	56.11	−0.685	Nuclear
	Dlo011228	*DlGATA10*	331	36,198.76	6.33	60.08	67.73	−0.518	Nuclear
	Dlo012990	*DlGATA12*	285	28,487.45	8.25	47.23	56.78	−0.728	Nuclear
	Dlo015565	*DlGATA15*	400	43,819.90	6.09	58.06	56.27	−0.672	Nuclear
	Dlo023262	*DlGATA17*	335	37,210.04	4.74	70.62	58.78	−0.671	Nuclear
	Dlo026193	*DlGATA20*	298	33,666.03	9.15	62.37	64.4	−0.687	Nuclear
	Dlo033619	*DlGATA22*	182	19,817.05	10.89	62.74	67.58	−0.513	Nuclear
III	Dlo007382	*DlGATA9*	300	32,575.97	6.56	39.46	61.70	−0.801	Nuclear
	Dlo013801	*DlGATA13*	287	31,263.42	6.15	40.09	57.39	−0.729	Nuclear
	Dlo013802	*DlGATA14*	344	37,626.45	4.62	48.36	66.02	−0.697	Nuclear
	Dlo023794	*DlGATA18*	354	39,276.64	4.99	49.87	60.34	−0.780	Nuclear
IV	Dlo000173	*DlGATA1*	317	35,755.20	9.59	61.84	52.11	−0.949	Nuclear
	Dlo001151	*DlGATA3*	192	21,530.58	9.84	46.67	55.36	−0.776	Nuclear
	Dlo003321	*DlGATA5*	168	18,519.80	9.58	58.8	67.38	−0.943	Nuclear
	Dlo012045	*DlGATA11*	112	12,336.41	9.93	55.23	61.07	−0.563	Nuclear
	Dlo015654	*DlGATA16*	275	30,480.31	8.32	66.27	34.11	−0.946	Nuclear

^1^ aa: amino acid number; ^2^ MW: molecular weight; ^3^ pI: isoelectric point; ^4^ GRAVY: grand average of hydropathicity.

## Data Availability

All analyzed data for this study are included in the contents of this article and Appendix A.

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
