# Peer review of "Genome-Wide Identification and Expression Analysis of GATA Family Genes in Dimocarpus longan Lour"

_ijms, 2024, doi:10.3390/ijms25020731_

Round 1
Reviewer 1 Report
Comments and Suggestions for Authors
In my opinion, this study on the genome-wide analysis and identification of the members of the DIGATA family in Dimocarpus longan Lour. is interesting. However, although the study seems to be well planned and carried out, some improvements need to be made before publication. The introduction should focus on the topic of the study without going into general issues. At a methodological level, there are issues that have not been included, such as the statistics shown in Figures 8 and 11, where the statistical differences observed under different conditions and stages of development are shown. More information on plant material and agronomic conditions should also be provided. At the results level, although well developed with 1 table and 11 figures, the possibility of reducing the number of figures is suggested. The format should be reviewed and an attempt should be made to place the figures (or tables) after they are referred to in the text. At the discussion level, I think it is correct, although there is always room for improvement by including more background, and the conclusions should be improved by stating the main findings in a global way. The rest of the comments are listed in the attached document.

In my opinion, minor editing of English language is required.
Reviewer 2 Report
Comments and Suggestions for Authors
The manuscript titled (Genome-Wide Identification and Expression Analysis of GATA Family Genes in Dimocarpus longan Lour) by Zheng et al. analyzed and identified members of the DlGATA family at the genome-wide level; this was mainly for the identification of DlGATA genes and their basic physicochemical properties, chromosomal localization, phylogenetic tree, gene structure, cis-regulatory elements, and expression patterns. In addition, the effects of treatments with the exogenous plant hormones IAA and ABA on the expression of GATA genes in early embryonic callus tissues of longan were also analyzed using real-time quantitative PCR, providing a theoretical basis for further research on the DlGATA transcription factor family.
The manuscript is generally well-written and structured. The analysis was successful, and the data was well understood and modeled in detail. In addition, the manuscript contains relevant paragraphs that have been discussed. The selection of the bibliography is appropriate to the content of the manuscript.
However, some errors appeared throughout the manuscript, making it difficult to accept it in its current version.
- According to the results the drought response mechanism of longan GATA is less associated with ABA, while there is a strong link with IAA. Based on this, it is possible to predict the role and function of DlGATA in the drought resistance process. My question is why the author doesn’t examine this hypothesis by doing a drought experiment to study the expression analysis of the studied genes to confirm this hypothesis. In my opinion, this experiment is essential.
- All Gene names in the manuscript must be italicized as TaGATAs.
- Please improve all the figures! All the resolution of the figures needs to improve intensely. Each figure should be self-explanatory! Clearly use description in the figure, title, or legend. Ensure they contain all necessary information so that a person can look at a figure and understand what all the components represent (including statistics, significance – explain whether the data show mean ±SD or mean ± SE, explain letters, test, p or α value etc.
- All the scientific names in the references must be in italics and need revision.
- Table 1 in non-editable format; kindly provide 'Table 4' in editable format.
- Figure 3 panels (A, B, C, D) must be cited in the results separately.
- Figure 5 must be transferred before section 2.5.
- Figure 7 must be transferred after section 2.6.
- Figure 10 must be transferred after section 2.8.
- I don’t think the S1 and S2 tables are of significant importance; they represent those same results in Figures 8 and 11, so deleting them from the MS is recommended.
Best Regards
Round 2
Reviewer 2 Report
Comments and Suggestions for Authors